# The Depositional Environment of the Lacustrine Source Rocks in the Eocene Middle Number of the Liushagang Formation of the Weixinan Sag, Beibuwan Basin, China: Implications from Organic Geochemical Analyses

**Xiaoyong Yang** [1,2]**, Xiaoxia Lv** [1,2,]*****, Yahao Huang** [3,]*****, Yunlong He** [1] **, Rui Yang** [4] **, Ruyue Wang** [5] **and Peng Peng** [6]

[1] Hubei Key Laboratory of Marine Geological Resource, China University of Geosciences, Wuhan 430074, China; ylhe@cug.edu.cn (Y.H.)
[2] State Key Laboratory of Biogeology and Environmental Geology, China University of Geosciences, Wuhan 430074, China
[3] College of Resource and Environment, Yangtze University, Wuhan 430100, China
[4] School of Earth Resources, China University of Geosciences, Wuhan 430074, China
[5] SINOPEC Petroleum Exploration and Production Research Institute, Beijing 102206, China
[6] PetroChina Tarim Oilfield Company, Korla 841000, China
***** Correspondence: lvxx@cug.edu.cn (X.L.); hyhtr08916@cug.edu.cn (Y.H.)

**Abstract:** The Eocene middle number of the Liushagang Formation (LS2) of the Weixinan Sag, Beibuwan Basin, characterized by a thick succession of excellent quality source rocks, is composed of lacustrine organic-rich shales, mudstones, and shales (mudstones/shales). However, the complex and specific depositional environment in the source rocks of LS2 raise questions about the mainly controlling factors of lacustrine organic matter (OM) accumulation. In this study, total organic carbon (TOC) contents, Rock-Eval pyrolysis, as well as biomarker data are used to investigate the nature of the depositional environment and the enrichment mechanism of OM in the source rocks of LS2. The values of $T_{max}$, CPI, $C_{29}$ steranes $\alpha\beta\beta/(\alpha\alpha\alpha+\alpha\beta\beta)$, and the 22S/(22S+22R) ratios of the $17\alpha$, $21\beta(H)$-$C_{31}$ hopane together confirm that the OM in the source rocks of LS2 is immature to of low maturity, which suggests that the nature of biomarkers may not be affected by thermal maturity. The hydrocarbon potential was higher in the organic-rich shales (with a mean of 20.99 mg/g) than in the mudstones/shales (with a mean of 7.10 mg/g). The OM in organic-rich shales is type I and II kerogen and that in mudstones/shales is type II kerogen. The $C_{27}/C_{29}$ regular steranes ratios and 4-methylsterane indices (4MSI) further confirmed the difference in the source of OM between organic-rich shales and mudstones/shales; that is, that the OM of organic-rich shales is mainly derived from the lake algae and aquatic macrophytes and the OM of mudstones/shales is mainly from the higher plants. The values of the gammacerane index and ratios of $C_{21}/C_{23}$ TT and $C_{24}$ Tet/$C_{26}$ TT all indicate that the source rocks from LS2 are deposited in freshwater to a low salinity water column. Moreover, a cross-plot of $C_{21-22}/C_{27-29}$ sterane versus dia/reg $C_{27}$ sterane ratios and Pr/Ph ratios suggests that the source rocks from LS2 are recorded to have sub-oxic to oxic conditions. Based on those analyses, two dynamical formation models were proposed: a high-productivity and oxic-suboxic dynamical formation model (Model A) and a low-productivity and oxic-suboxic dynamical formation model (Model B).

**Keywords:** Beibuwan Basin; Weixinan Sag; organic-rich shales; biomarker; depositional environment; enrichment of organic matter

## 1. Introduction

Organic-rich fine-grained sediments are significant hydrocarbon source rocks and, in recent years, have become key targets for shale oil exploration and production in China [1]. The Liushagang Formation, widely developed in the Weixinan Sag, Beibuwan Basin, was

considered to be excellent source rocks and a shale oil target interval in the northwestern South China sea [2–10].

In recent years, the massive amounts of research into organic-rich shales were concentrated in the Liushagang Formation in the Weixinan Sag, Beibuwan Basin [5–9]. Analyzing the characteristics of organic geochemistry and geobiology in the whole Liushagang Formation, Ye et al. (2020) proposed that the paleo-productivity, organic matter burial efficiency, and organic carbon burial productivity of the source rocks were the significant factors limiting the development of excellent quality source rocks. The hyper-trophic lake is developed during the LS2 sedimentary period, which indicated that hydrogen-rich components might be derived from the endogenous aquatic algae of lakes, providing most of the hydrocarbon potential of excellent quality source rocks [5].

Paradoxically, Xu et al. (2021) proposed that hydrogen-rich components including liptinite, cutinite, and sporinite might be from terrigenous higher plants, providing most of the hydrocarbon potential of excellent quality source rocks and oil shales. Samples of general quality and excellent quality source rocks and oil shales from the whole Liushagang Formation in the Weixinan Sag, Beibuwan Basin were analyzed to investigate their organic geochemistry, palynofacies, and trace elements, and it was further pointed out that fresh-brackish and weak oxidation reducing conditions were the main controlling factors for excellent quality source rocks [6].

In addition, Cao et al. (2020) investigated the mechanisms for the accumulation of organic matter in source rocks from LS2 of Weixinan Sag by analyzing the depositional environment and lithological characteristics of the parent rocks using elemental geochemistry and biomarker data, which pointed out that the inter-relationships among algal blooms, paleoproductivity, and paleoredox conditions played a significant role for excellent quality source rocks [7–9].

Therefore, the nature of the conditions and accumulation mechanism of organic matter in LS2 of Weixinan Sag were unclear. In conclusion, detailed investigation into environment changes and the enrichment mechanism of OM during the period of LS2 sedimentary play a significant role in understanding their nature.

Here, we generated a number of total organic carbon (TOC) content, Rock-Eval pyrolysis, as well as biomarker data from the drilling rocks analyses in LS2 located in Weixinan Sag, Beibuwan Basin. The objectives are to (1) reconstruct the nature of the paleo-environment that occurs in the deposition of the source rocks from LS2 in Weixinan Sag by assessing the degree of paleo-productivity, redox conditions and water salinity; (2) analyze the mechanisms responsible for the accumulation of organic matter from two source rocks groups, organic-rich shales and mudstones/shales, in LS2, respectively; (3) based on those analyses, two formation models of the lacustrine OM accumulation in the source rocks of LS2 are presented.

## 2. Geologic Setting and Samples

The Beibuwan basin, a Cenozoic extensional one, is located in the northern continental shelf of the South China Sea, covering an area of $3.9 \times 10^4$ km$^2$, where a number of oil and gas fields have been explored, mainly in Weishan Sag, Wushi Sag, and Fushan Sag [11]. The Weixinan Sag, a third-level structural unit, spanning an area of about 3000 km$^2$, is located in the northwest of the Beibuwan Basin, which is a dustpan-shaped fault depression caused by Shenhu Movement in the late period of the Paleocene [12]. Based on the configuration of this depression, the evolution of the Weixinan Sag experienced syn-rift lacustrine depositions during the Paleogene, followed by the deposition of marine sediments from the Neogene to present (Figures 1 and 2) [13,14].

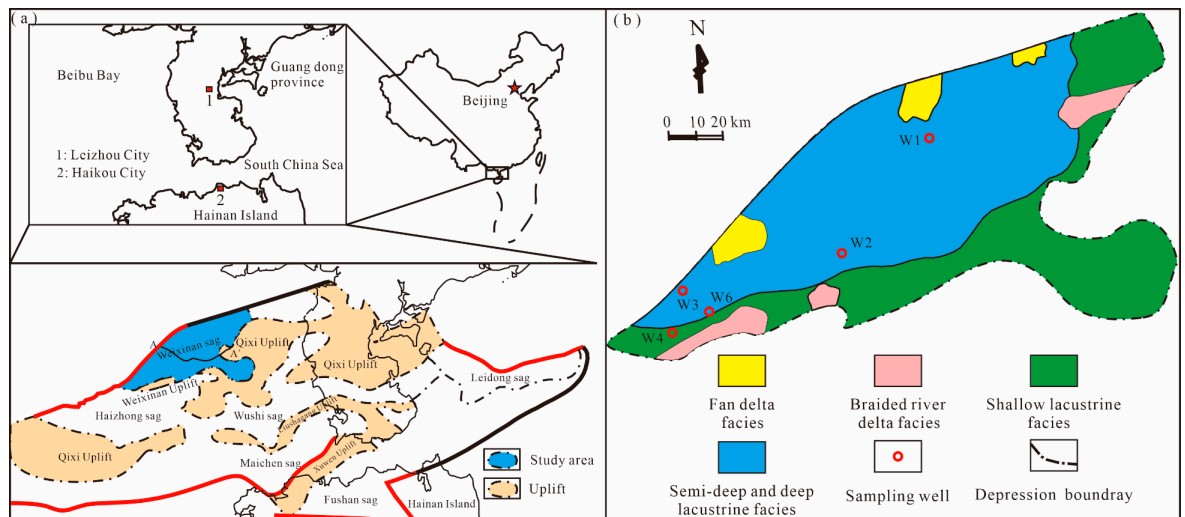

**Figure 1.** Location of the study area (**a**), depositional facies of LS2 (**b**), and the locations of sampling wells in Weixinan Sag (**b**) from Beibuwan Basin, South China Sea (modified from Refs. [4,9].

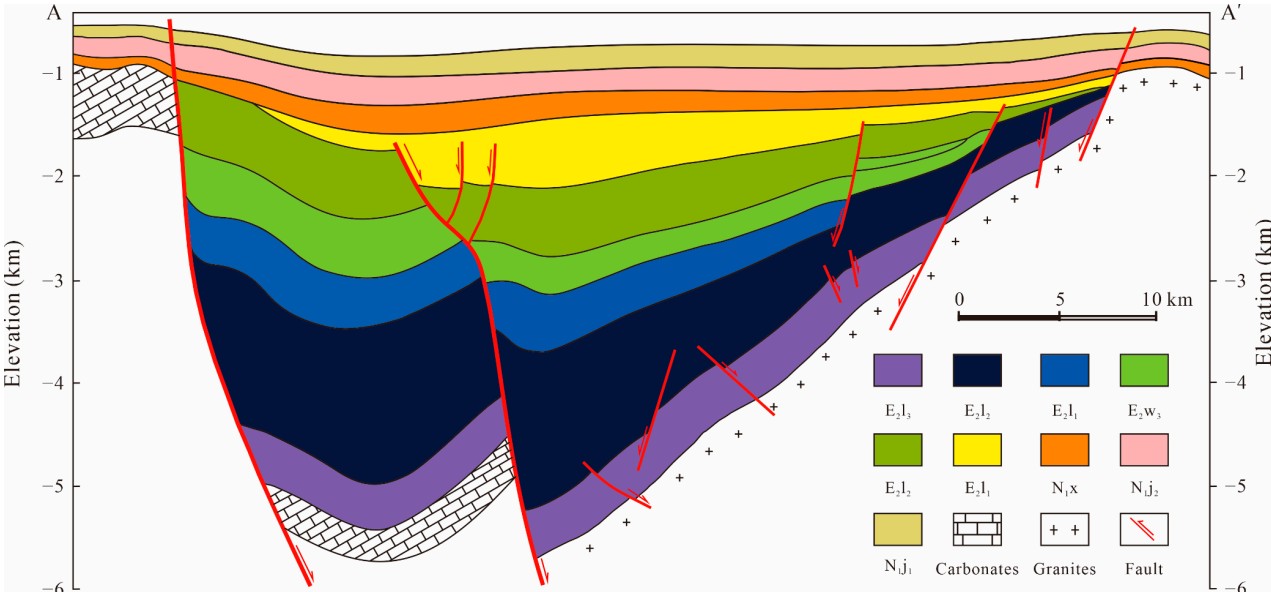

**Figure 2.** Cross section showing the structural framework of Weixinan Sag from Beibuwan Basin, South China Sea (modified from Ref. [10]. Section locations are shown in Figure 1.

The Cenozoic strata of the sag, from bottom to top, are composed of Paleogene Changliu, Liushagang, Weizhou Formation, Neogene Xiayang, Jiaowei, Dengloujiao, Wanglougang Formation, and Quaternary strata, among which the Eocene Liushagang Formation is the main source rock of the sag, which can be further subdivided into the top number (LS1), middle number (LS2), and bottom number (LS3). In addition, based on the depositional characteristics of the Liushagang Formation, Weixinan Sag can be classified into A, B, C, and D sub-depressions (Figure 3) [5].

During the deposition period of LS2, basin subsidence further intensified, resulting in the development of the sag reaching its peak. Controlled by these characteristics, the sedimentary facies of LS2 are mainly semi-deep lacustrine-deep lacustrine facies and the lithology mainly is composed of shales and mudstones [15,16].

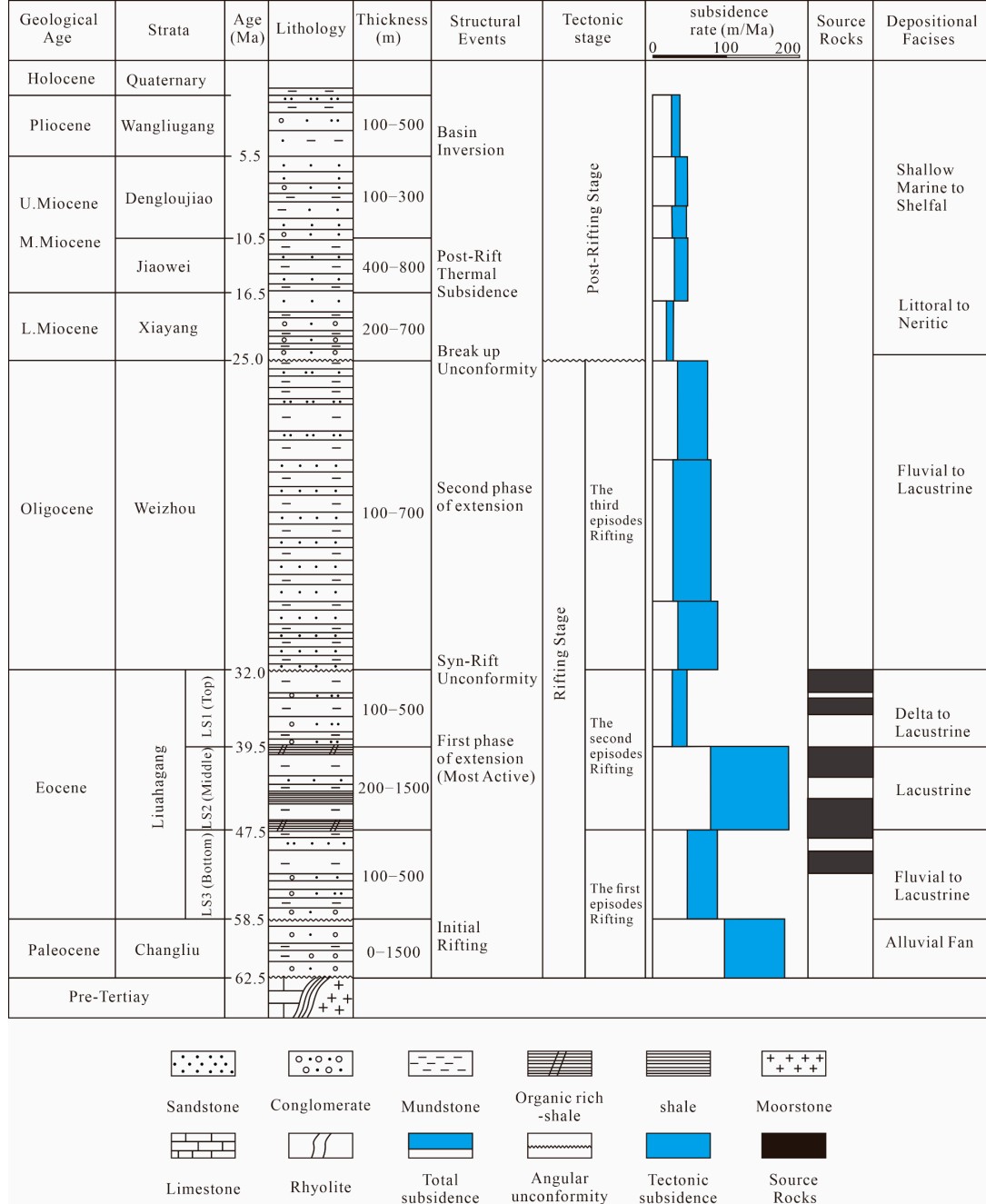

**Figure 3.** Schematic view showing the stratigraphic column of the Beibuwan Basin (modified from Refs. [7,9].

The lacustrine shale of LS2 is mainly composed of gray-black shale and gray-black calcareous shale, developed for horizontal bedding and intermittent horizontal bedding [9,17]. The total organic carbon content of the lacustrine shale is generally high, with most of them classified as organic-rich shales, indicating that the water column is calm during deposition without benthic disturbance and a relatively high deposition rate. Visible banded pyrite particles and a high content of reduced sulfur are found in the lacustrine shale, which suggest that the sediments may be anoxic during deposition [17].

Moreover, the relative abundances of clay minerals of lacustrine shale from LS2 follow the order of illite > illite/smectite mixed-layers > kaolinite > smectite > chlorite, which may indicate a relatively warm and humid climate environment during deposition in that period [17].

Here, 31 drilling rock samples, including 15 organic-rich shales, 11 mudstones, and 5 shales (the definition is from 4.1) were collected from LS2 in well 1 (4 samples), 2 (7 samples), 3 (7 samples), 4 (9 samples), and 6 (4 samples) in the Weixinan Sag, Beibuwan Basin.

## 3. Analytical Methods

### 3.1. TOC Contents

The TOC contents of the collected source rock samples were analyzed with a LECO-230 carbon-sulfur analyzer (CS-230, Laboratory Equipment Corporation (LECO), San Joseph, MI, USA). The experiments were carried out at a temperature of 25 °C and a relative humidity of 40%. Approximately 120–200 mg of powdered rock samples, less than 100 mesh, was added to hydrochloric acid (10%) at a consistent temperature of 60 °C to 80 °C until no reaction occurred to remove carbonate, and then used deionized water to rinse the residual hydrochloric acid. Finally, these powder samples were freeze-dried to measure the TOC content. Based on replicate analyses and standards, the analytical errors of TOC content were better than $\pm 0.2\%$.

### 3.2. Rock-Eval Pyrolysis

Rock-Eval pyrolysis was performed using a Rock-Eval 6 instrument (Vinci Technologies, Nanterre, France) with a flame ionization detector. Each of the crushed source rock samples, approximately 80–120 mg, were initially rapidly heated to 300 °C, then held at 300 °C for 1 min, then heated from 300 °C to 600 °C at 50 °C/min, and finally held at 600 °C for 8 min. The hydrocarbon generation evaluation parameters measured by these methods included $S_1$ (mg HC/g rock), $S_2$ (mg HC/g rock), $S_3$ (mg $CO_2$/g rock), and $T_{max}$ (the temperature of maximum pyrolysis yield, °C).

### 3.3. Extraction and Separation

Aliquots of 10–40 g of powdered rock, less than 100 mesh, were extracted in Soxhlet apparatus using DCM and MeOH (9:1 by vol.) at a constant temperature of 55 °C for approximately 72 h. The asphaltene was precipitated from the total extract samples using an excessive volume of n-hexane overnight in a freezer. After removing the asphaltene, the resultant maltene was separated through a silica gel/alumina column, using n-hexane, n-hexane/dichloromethane (1:2 by vol.), and dichloromethane/methanol (98:2 by vol.) sequentially to elute saturated, aromatic, and resin fractions.

### 3.4. Instrumental Analyses

3.4.1. GC Analysis

These saturate fractions were analyzed using gas chromatography (GC) on a HP6890 instrument equipped with fused silica capillary column (30 m × 0.25 mm HP-5). The oven temperature for saturate fractions in the GC analysis was initially held at 40 °C for 5 min, then raised to 80 °C at 2 °C/min, then from 80 to 300 °C with 4 °C/min, and held for 30 min at 300 °C as the final temperature. Helium, at 1.5 mL/min, was used to carry gas and data were obtained using an HP ChemStation.

3.4.2. GC-MS Analysis

These saturated and aromatic fractions were immediately sealed and then conducted using HP6890GC/5973MSD (ionization energy was 70 eV) equipped with an HP-5MS fused silica column (30 m × 0.25 mm, film thickness 0.25 μm). A mass selective detector (MSD) was operated in full scan acquisition mode and the scan range was about 50~550 amu. The GC oven temperature for analysis of the saturated fractions was initially held at 50 °C for 1 min and then ramped from 50 °C to 100 °C at 20 °C/min, from 100 °C to 315 °C at 3 °C/min, and eventually held at 315 °C for 16 min. The temperature of injector was 300 °C. Helium, with a flow rate of 1.0 mL/min, was used to carry gas. The temperature program for the aromatic fraction was as follows: oven temperature was initially held at 60 °C for

2 min, then programmed to 150 °C at 8 °C/min, to 320 °C at 4 °C/min, and finally held at 320 °C for 10 min. The injector temperature was 280 °C. Helium was used as a carrier gas at 1.4 mL/min. The relative abundance of molecular markers was determined from peak areas in the relevant mass chromatograms.

## 4. Results

### 4.1. Bulk Organic Matter Compositions and Properties

The total organic carbon (TOC) contents and values of hydrogen index (HI, calculated from $100 \times S_2/\text{TOC}$) for source rock samples from the LS2 range from 1.12 to 5.70 wt% and from 143 to 707 mg HC/g TOC, respectively (Table 1). Variations in HI and TOC of the studied samples are well correlated, and the values of pyrolytic yield (calculated from $S_1 + S_2$), similarly, range between 1.64 and 38.53 mg/g. Their average values are 2.79 wt%, 422 mg HC/g TOC, and 13.82 mg/g, respectively (Table 1).

**Table 1.** Basic geochemical data for source rock samples from LS2 of the Weixinan Sag, Beibuwan Basin.

| Wells | Depth (m) | Lithology | Number | TOC (wt%) | HI (mg HC/gTOC) | Tmax (°C) | (S1 + S2) (mg/g) |
|-------|-----------|-----------|--------|-----------|------------------|-----------|------------------|
| W1 | 1635–1650 | Organic-rich shale | LS2 | 4.69 | 634 | 427 | 30.22 |
| W1 | 1716.39–1717.39 | Mudstone | LS2 | 1.83 | 399 | 432 | 7.35 |
| W1 | 1770–1780 | shale | LS2 | 2.02 | 332 | 431 | 6.78 |
| W1 | 1830–1835 | Mudstone | LS2 | 1.34 | 240 | 437 | 3.26 |
| W2 | 1954–1960 | Mudstone | LS2 | 2.2 | 502 | 427 | 11.22 |
| W2 | 2015–2025 | Organic-rich shale | LS2 | 4.42 | 707 | 432 | 31.74 |
| W2 | 2100–2130 | shale | LS2 | 1.85 | 439 | 434 | 8.24 |
| W2 | 2186–2192 | Mudstone | LS2 | 2 | 416 | 433 | 8.45 |
| W2 | 2226–2234 | Organic-rich shale | LS2 | 5.38 | 614 | 431 | 33.57 |
| W2 | 2249–2280 | Organic-rich shale | LS2 | 5.53 | 643 | 432 | 36.22 |
| W2 | 2286–2290 | Organic-rich shale | LS2 | 5.7 | 663 | 433 | 38.53 |
| W3 | 3098–3122 | shale | LS2 | 2.17 | 370 | 433 | 8.54 |
| W3 | 3168–3190 | Mudstone | LS2 | 1.63 | 339 | 437 | 5.84 |
| W3 | 3213–3235 | Organic-rich shale | LS2 | 2.34 | 310 | 432 | 8.20 |
| W3 | 3250–3262 | Mudstone | LS2 | 2.11 | 316 | 437 | 7.40 |
| W3 | 3332–3340 | Organic-rich shale | LS2 | 4.4 | 531 | 437 | 26.08 |
| W3 | 3361.3–3363 | Organic-rich shale | LS2 | 2.92 | 446 | 441 | 14.77 |
| W3 | 3375–3385 | Organic-rich shale | LS2 | 2.24 | 387 | 439 | 10.00 |
| W4 | 3010–3020 | Mudstone | LS2 | 1.56 | 273 | 435 | 4.38 |
| W4 | 3072–3082 | Organic-rich shale | LS2 | 2.24 | 339 | 427 | 7.80 |
| W4 | 3130–3140 | Mudstone | LS2 | 2.19 | 364 | 427 | 8.16 |
| W4 | 3200–3206 | Organic-rich shale | LS2 | 4.95 | 604 | 436 | 30.94 |
| W4 | 3250–3260 | shale | LS2 | 2.19 | 468 | 438 | 10.60 |
| W4 | 3312–3322 | Mudstone | LS2 | 2.15 | 378 | 435 | 8.36 |
| W4 | 3383–3396 | Organic-rich shale | LS2 | 2.46 | 355 | 433 | 8.94 |
| W4 | 3448–3460 | Organic-rich shale | LS2 | 3.84 | 482 | 436 | 19.21 |
| W4 | 3490–3496 | Mudstone | LS2 | 1.93 | 375 | 433 | 7.43 |
| W6 | 2262–2274 | Mudstone | LS2 | 1.12 | 143 | 427 | 1.64 |
| W6 | 2292–2302 | shale | LS2 | 1.80 | 323 | 424 | 5.92 |
| W6 | 2330–2340 | Organic-rich shale | LS2 | 2.22 | 305 | 421 | 6.95 |
| W6 | 2368–2378 | Organic-rich shale | LS2 | 2.96 | 391 | 427 | 11.75 |

Integrated the logging data and rock observations, based on the different distributions of TOC, the source rocks samples from LS2 can be divided into organic-rich shales and general mudstones and shales (mudstones/shales). The TOC contents of the organic-rich shales, with a mean of 3.75 wt%, range from 2.22 to 5.70 wt%, and those of the mudstones/shales vary from 1.12 wt% to 2.20 wt%, with an average of 1.88 wt% (Table 1). In addition, the HI and pyrolytic yield of organic-rich shales range from 305 to 707 mg HC/g TOC and 6.95 to 38.53 mg/g, with a mean of 494 mg HC/g TOC and 20.99 mg/g, respectively. Those of general mudstones/shales vary from 143 to 502 mg HC/g TOC and 1.64 to 11.22 mg/g, respectively. Their average values are 355 mg HC/g TOC and 7.10 mg/g, respectively.

### 4.2. Biomarkers

#### 4.2.1. *n*-alkanes and the Pr/Ph Ratio

*n*-alkanes of studied source rock samples are relatively abundant, from which *n*-$C_{12}$ to *n*-$C_{38}$ were detected (Figure 4). TAR, ratios of certain *n*-alkanes, and values in organic-rich shales range from 0.25 to 1.82 with a mean of 0.98, and those in mudstones/shales range from to 0.18 to 3.25, with an average value of 1.37.

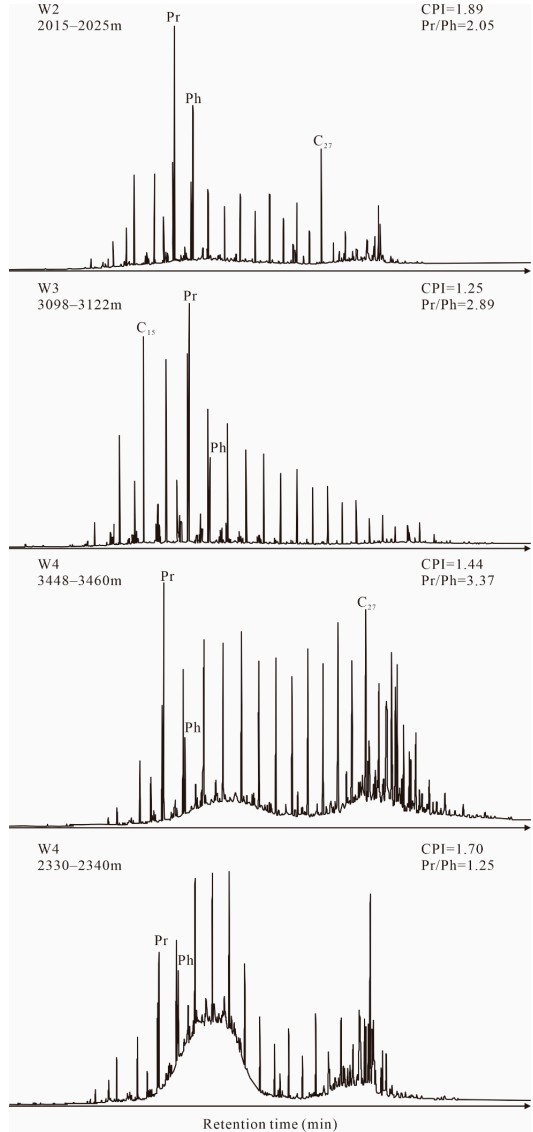

**Figure 4.** Gas chromatograms of the extractable saturated fractions showing the distribution of *n*-alkanes for source rock samples from LS2 of the Weixinan Sag, Beibuwan Basin.

The CPI (carbon preference index) values of the studied source rock samples range from 1.11 to 2.69, with a mean of 1.49, which indicate that the *n*-alkanes exhibit an obvious odd predominance (Table 2). The values of Pr/Ph ratios range from 1.25 to 3.37, with an average value of 2.37 (Table 2). Those in organic-rich shales vary from 1.25 to 3.37, with a mean of 2.54, and those in mudstones/shales range from 1.42 to 3.07, with a mean of 2.14 (Table 2).

#### 4.2.2. Tricyclic Terpane

The tricyclic terpane, with carbon numbers ranging from 19 to 29, of these studied samples has the main distribution pattern: near normal distribution with a maximum peak

at $C_{21}$ (Figure 5). The values of $C_{21}/C_{23}$ TT and $C_{24}$ Tet/$C_{26}$ TT ratios in these source rock samples range from 0.86 to 3.00 and 0.97 to 2.42, with a mean of 1.67 and 1.58, respectively (Table 2). The value of $C_{21}/C_{23}$ TT ratios in the organic-rich shales and shales/mudstones varied from 0.86 to 3.00 with an average of 1.59, and from 0.87 to 2.98 with a mean of 1.77, respectively (Table 2). The values of $C_{24}$ Tet/$C_{26}$ TT ratios of the organic-rich shales were between 1.20 and 1.95 with a mean of 1.54, and those of the shales/mudstones were from 0.97 to 2.42, with an average of 1.62 (Table 2).

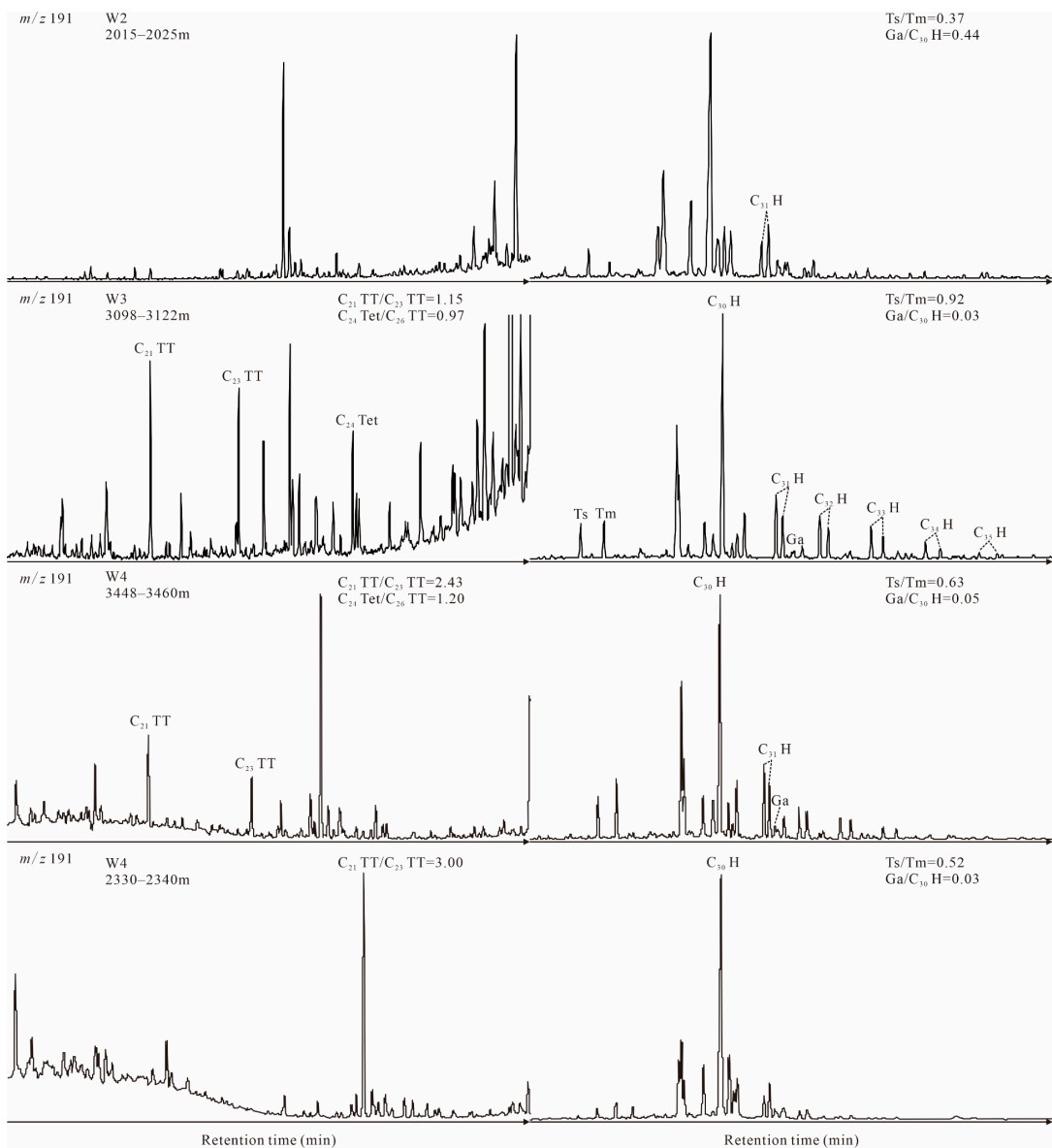

**Figure 5.** Selected m/z 191 mass fragmentograms for source rock samples from LS2 of the Weixinan Sag, Beibuwan Basin, showing the distribution of the terpane series.

### 4.2.3. Hopanes and Gammacerane

The distribution of hopanes in the studied samples is characterized by the presence of 17α,21β(H)- and 17β,21α(H)-hopanes, ranging from $C_{27}$ to $C_{35}$, with 17α,21β-22R $C_{30}$ hopane being the most abundant one (Figure 5). Gammacerane was also found in all samples, with the values of gammacerane index (GI, calculated as gammacerane/$C_{30}$ αβ-hopane) ranging from 0.03 to 0.47, the average value of which was 0.11. The values of GI of the organic-rich shales were between 0.03 and 0.44 with a mean of 0.10, and those of the shales/mudstones were from 0.03 to 0.47, with an average of 0.12 (Table 2).

**Table 2.** Selected molecular parameters for source rock samples from LS2 of the Weixinan Sag, Beibuwan Basin.

| Wells | Depth (m) | Lithology | Number | Pr/Ph | CPI | TAR | $C_{21}/C_{23}$ TT | $C_{24}$ Tet/$C_{26}$ TT | G/$C_{30}$ H | $C_{21+22}/C_{27-29}$ S | Dia/reg $C_{27}$ S | $C_{27}/C_{29}$ S | 4MSI |
|---|---|---|---|---|---|---|---|---|---|---|---|---|---|
| W1 | 1635–1650 | Organic-rich shale | LS2 | / | / | / | / | / | 0.03 | 0.00 | 0.00 | 0.94 | 0.00 |
| W1 | 1770–1780 | shale | LS2 | / | / | / | / | / | 0.09 | 0.00 | 0.31 | 0.80 | 0.00 |
| W1 | 1830–1835 | Mudstone | LS2 | / | / | / | / | / | 0.06 | 0.00 | 0.28 | 0.70 | 0.00 |
| W1 | 1716.39–1717.39 | Mudstone | LS2 | / | / | / | / | / | 0.13 | 0.00 | 0.21 | 0.87 | 0.00 |
| W2 | 1954–1960 | Mudstone | LS2 | 1.62 | 1.74 | 0.36 | / | / | 0.30 | 0.00 | 0.23 | 0.88 | 0.00 |
| W2 | 2015–2025 | Organic-rich shale | LS2 | 2.05 | 1.89 | 0.66 | / | / | 0.44 | 0.00 | 0.26 | 0.95 | 0.00 |
| W2 | 2100–2130 | shale | LS2 | 1.81 | 1.84 | 0.76 | / | / | 0.47 | 0.00 | 0.57 | 0.90 | 0.00 |
| W2 | 2249–2280 | Organic-rich shale | LS2 | 3.02 | 1.63 | 0.71 | / | / | 0.29 | 0.00 | 0.43 | 1.06 | 0.00 |
| W2 | 2286–2290 | Organic-rich shale | LS2 | 2.93 | 1.13 | 1.77 | / | / | 0.05 | 0.00 | 0.57 | 1.02 | 0.00 |
| W3 | 3098–3122 | shale | LS2 | 2.89 | 1.25 | 0.18 | 1.15 | 0.97 | 0.03 | 0.03 | 0.65 | 0.78 | 0.42 |
| W3 | 3213–3235 | Organic-rich shale | LS2 | 3.03 | 1.15 | 0.63 | 0.87 | 1.26 | 0.03 | 0.06 | 0.83 | 0.71 | 0.38 |
| W3 | 3250–3262 | Mudstone | LS2 | 2.90 | 1.22 | 0.65 | 0.87 | 1.36 | 0.05 | 0.03 | 0.68 | 0.77 | 0.31 |
| W3 | 3332–3340 | Organic-rich shale | LS2 | 2.81 | 1.12 | 0.43 | 0.94 | 1.58 | 0.03 | 0.05 | 0.65 | 0.63 | 0.34 |
| W3 | 3361.3–3363 | Organic-rich shale | LS2 | 2.73 | 1.11 | 0.25 | 0.90 | 1.56 | 0.06 | 0.05 | 0.74 | 0.56 | 0.32 |
| W3 | 3375–3385 | Organic-rich shale | LS2 | 2.55 | 1.17 | 1.29 | 0.86 | 1.69 | 0.08 | 0.06 | 0.73 | 0.64 | 0.25 |
| W4 | 3130–3140 | Mudstone | LS2 | 1.80 | 1.39 | 3.25 | 1.98 | 1.55 | 0.03 | 0.02 | 0.58 | 1.38 | 0.34 |
| W4 | 3200–3206 | Organic-rich shale | LS2 | 2.23 | 1.42 | 1.82 | 1.31 | 1.95 | 0.04 | 0.03 | 0.53 | 1.48 | 0.46 |
| W4 | 3250–3260 | shale | LS2 | 1.63 | 1.45 | 2.03 | 1.01 | 2.42 | 0.08 | 0.04 | 0.59 | 1.22 | 0.20 |
| W4 | 3448–3460 | Organic-rich shale | LS2 | 3.37 | 1.44 | 1.80 | 2.43 | 1.20 | 0.05 | 0.05 | 0.54 | 1.20 | 1.08 |
| W4 | 3490–3496 | Mudstone | LS2 | 3.07 | 1.42 | 2.55 | 2.98 | 1.81 | 0.07 | 0.05 | 0.65 | 1.12 | 0.74 |
| W6 | 2292–2302 | shale | LS2 | 1.42 | 2.69 | 1.16 | 2.64 | / | 0.05 | 0.00 | 1.57 | 0.71 | 1.38 |
| W6 | 2330–2340 | Organic-rich shale | LS2 | 1.25 | 1.70 | 0.73 | 3.00 | / | 0.03 | 0.00 | 0.18 | 2.18 | 1.50 |
| W6 | 2368–2378 | Organic-rich shale | LS2 | 1.96 | 1.51 | 0.68 | 2.38 | / | 0.06 | 0.02 | 0.40 | 0.91 | 0.38 |

Pr: pristane; Ph: phytane; TAR = $(C_{27} + C_{29} + C_{31})/(C_{15} + C_{17} + C_{19})$ *n*-alkanes; $C_{21}/C_{23}$ TT: $C_{21}/C_{23}$ tricyclic terpane; $C_{24}$ Tet/$C_{26}$ TT: $C_{24}$ 17,21-secohopane/$C_{26}$ tricyclic terpanes; Ga/$C_{30}$ H: Gammacerane/$C_{30}$ hopane; $C_{21+22}/C_{27-29}$ S: (pregnane + homopregnane)/$C_{27-29}$ regular steranes; 4MSI: 4-methylsterane/$C_{29}$ regular sterane.

### 4.2.4. Steranes

Steranes were detected in all samples, including regular steranes and diasteranes (Figure 6). The values of $C_{27}/C_{29}$ regular sterane ratios in the studied samples vary from 0.56 to 2.18, among which the values of $C_{27}/C_{29}$ regular steranes ratios of the organic-rich shales were between 0.056 and 2.18, with a mean of 1.00, and those of the shales/mudstones were from 0.70 to 1.38, with an average of 0.92 (Table 2).

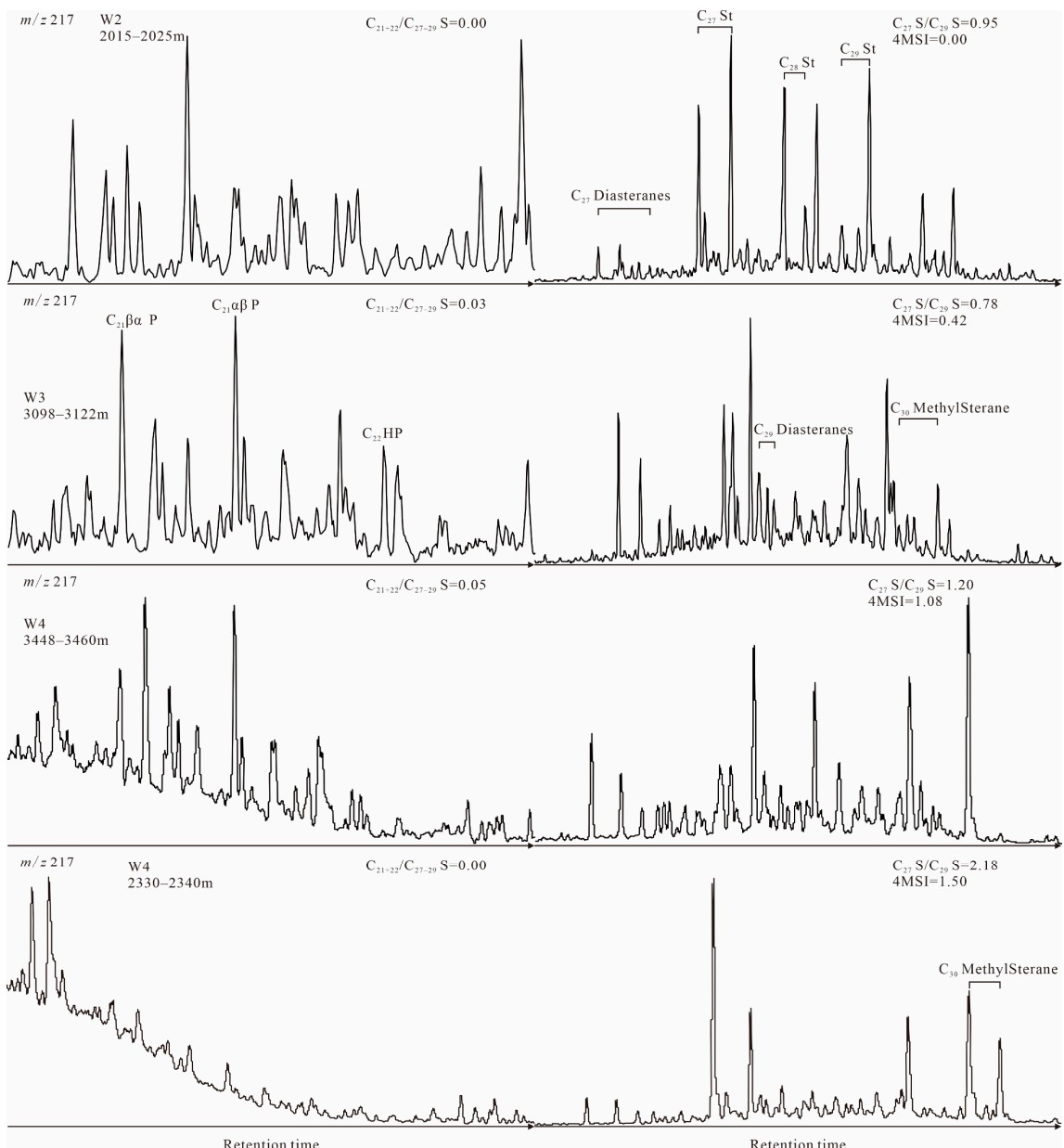

**Figure 6.** Selected m/z 217 mass fragmentograms for source rock samples from LS2 of the Weixinan Sag, Beibuwan Basin, showing the distribution of the sterane series.

Significantly, the values of $C_{27}$ diasteranes/$C_{27}$ regular steranes and $C_{21-22}/C_{27-29}$ steranes range from 0.00 to 1.57 and 0.00 to 0.06, respectively (Table 2).

The 4MSI (calculated from 4-methylsterane/$C_{29}$ regular sterane) in the organic-rich shales were between 0.00 and 1.50, with a mean of 0.39, and those of the shales/mudstones were from 0.00 to 1.38, with an average of 0.35, respectively (Table 2).

## 5. Discussion

### 5.1. Maturity Assessment

Relatively low values of $T_{max}$, ranging from 421 to 441 °C, indicate that the OM of the studied source rock samples is of low maturity to mature in LS2, and the maturity is similar [18,19]. The *n*-alkanes exhibiting an obvious odd predominance also confirms this conclusion.

Previous systematic studies have found that the $C_{29}$ steranes $\alpha\beta\beta/(\alpha\alpha\alpha+\alpha\beta\beta)$ and the 22S/(22S+22R) ratios of the $17\alpha,21\beta(H)$-$C_{31}$ hopanes can be effectively used to assess the maturity of OM [20]. The values of these ratios are distributed in both mature and low maturity areas; the highest maturity of these samples is only near the oil-generating window, which suggests that the nature of the biomarker may not be affected by thermal maturity (Figure 7).

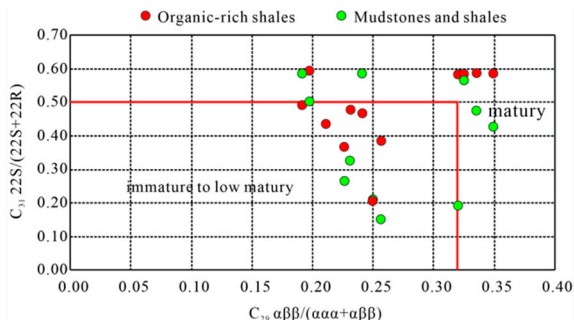

**Figure 7.** Cross-plot of $C_{29}\alpha\beta\beta/(\alpha\alpha\alpha+\alpha\beta\beta)$ sterane versus $C_{31}$22S/(22S+22R) hopane (modified from Ref. [21]).

### 5.2. Sources of Organic Matter

In this study, based on the cross-plot of HI versus $T_{max}$ values showing the organic matter types for source rock samples, the OM of organic-rich shales is type I and II kerogen and that of mudstones/shales is type II kerogen (Figure 8).

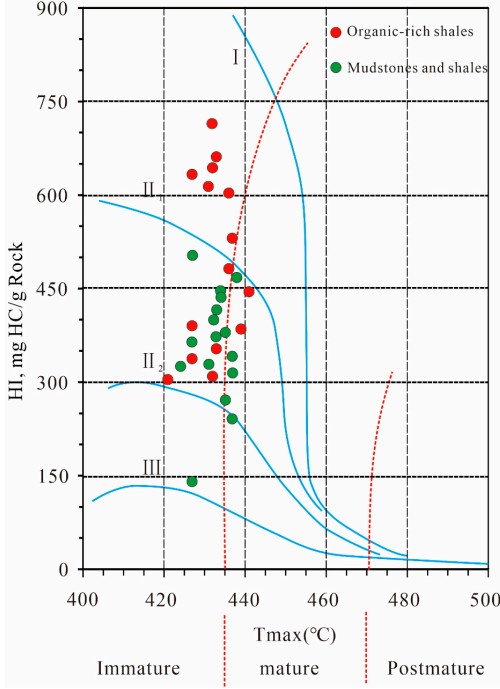

**Figure 8.** Cross-plot of HI vs. $T_{max}$ values showing the organic matter types for source rock samples from LS2 of the Weixinan Sag, Beibuwan Basin (I: type I kerogen; $II_1$: type $II_1$ kerogen; $II_2$: type $II_2$ kerogen; III: type III kerogen).

Ratios of certain *n*-alkanes, for example, TAR (equation annotated in Table 2), can be applied to indicate variations in the relative amounts of terrigenous versus aquatic hydrocarbons in sediment or rock extracts [22]. Higher ratios (TARs) indicate more terrigenous input from the surrounding watershed relative to aquatic sources [22]. In this study, the mean of value of TAR ratios in organic-rich shales is much lower than that in mudstones/shales, which indicates more terrigenous inputting from the surrounding watershed relative to aquatic sources in mudstones/shales and more aquatic sources relative to terrigenous inputting in organic-rich shales.

The plot formed by the ratios of pristane to $C_{17}$ *n*-alkanes and phytane to $nC_{18}$ *n*-alkanes ($Pr/nC_{17}$ and $Ph/nC_{18}$) can also be effectively applied to the organic matter types, depositional environment, and maturity of source rocks. As shown in Figure 9, the organic matter of source rocks from LS2 is mixed [23].

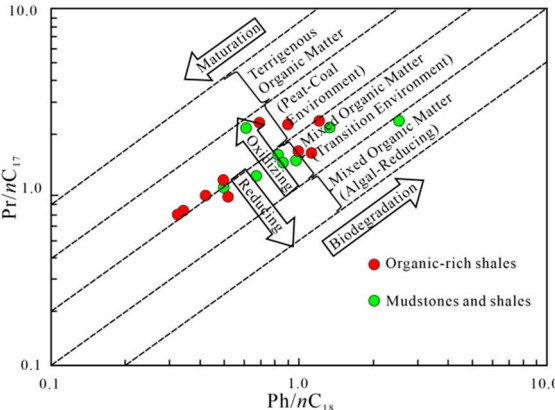

**Figure 9.** Cross-plot of $Pr/nC_{17}$ vs. $Ph/nC_{18}$ ratios for source rock samples from LS2 of the Weixinan Sag, Beibuwan Basin (modified from Ref. [23]).

Extensive studies showed that the abundance of $C_{27}$ regular steranes relative to $C_{29}$ regular steranes can also be used to reflect the biological resource of the source rock [24,25]. The values of $C_{27}/C_{29}$ regular steranes ratios of organic-rich shales from LS2 range from 0.56 to 2.18, with a mean of 1.02, which suggest that the organic matter of these samples is mainly from the lake algae and aquatic macrophytes, whereas those values of mudstones/shales vary from 0.70 to 1.38, with an average of 0.35, indicating that their organic matter is mainly derived from terrestrial higher plants.

### 5.3. Plaeoproductivity

It has been suggested that $C_{30}$-4-methyl steranes have two biological sources, one occurring from natural products of dinoflagellates and the other derived from bacteria (methanogens, methylococcus capsulatus) [26–29].

The abundance of $C_{30}$-4-methylsterane is found in samples from LS2. Owing to the special and unique depositional environment of LS2 of the Weixinan Sag, Beibuwan Basin, the biological sources of $C_{30}$-4-methylsterane may only be derived from natural products of dinoflagellates, which may indirectly reflect the level of algal biomass.

In this study, significantly, the values of the 4MSI in organic-rich shales from LS2 were relatively higher than those of the mudstones/shales, which suggests that the paleo-lake experienced more severe algal blooms during the sedimentation of the organic-rich shales compared with during the deposition of the mudstones/shales, and the results are similar to those of Cao et al., 2020 [7].

### 5.4. Paleoenvironmental Changes
#### 5.4.1. Redox Conditions

The isoprenoids pristane (Pr) and phytane (Ph) are found in all of the studied samples (Figure 5). Previous systematic studies have shown that high Pr/Ph (>3.0) ratios suggest

oxic conditions, while low values (<0.8) indicate an anoxic environment [30]. The values of Pr/Ph ratios in samples from LS2 range between 1.25 and 3.37, with a mean of 2.37. Moreover, all samples were determined to be of low maturity, which indicated that the nature of Pr/Ph ratios may not be affected by variability in thermal maturity. Therefore, the relatively high values of Pr/Ph ratios observed in LS2 are likely to be a trustworthy recorder of sub-oxic to oxic conditions.

In addition, according to Wang et al. (2015), pregnane and its homologues are related t restricted, clastic-starved marine, or saline depositional settings [31]. Therefore, a cross-plot of $C_{21-22}/C_{27-29}$ sterane versus dia/reg $C_{27}$ sterane ratios was proposed to illustrate the diagnosis of depositional environments. As shown in Figure 10, the values of $C_{21-22}/C_{27-29}$ sterane and dia/reg $C_{27}$ sterane ratios of samples from LS2 all fall into the area of a relatively open environment.

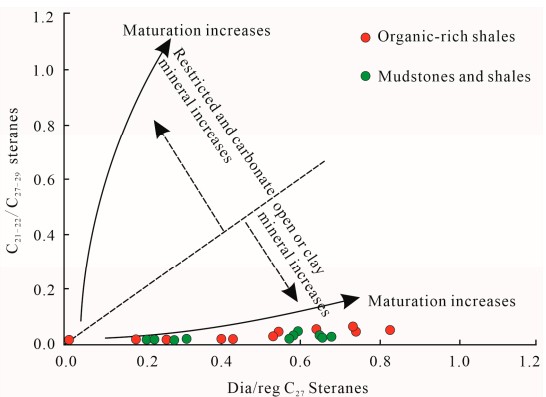

**Figure 10.** Cross-plot of $C_{21-22}/C_{27-29}$ sterane versus dia/reg $C_{27}$ sterane ratios (modified from Ref. [31]).

### 5.4.2. Water Salinity

Gammacerane, thanks to its special biological sources, is widely regarded as being highly specific for water column stratification (commonly owing to hyper-salinity) during the deposition of sediments. Therefore, when the water column has a higher salinity, the stratification of the water column will be more distinct, which may produce abundant gammacerane [32,33].

Significantly, the gammacerane index (GI) was proposed to reflect changes in salinity, which are calculated from the concentration of gammacerane relative to $C_{30}$ $\alpha\beta$-hopane [32,33]. In this study, the observed values of GI of all studied samples from LS2 range from 0.03 to 0.47, with a mean of 0.11, which indicates that they are deposited in a fresh water to brackish environment without water stratification.

In general, for the distribution patterns of $C_{19-26}$, the tricyclic terpane series of lacustrine source rocks presents a normal distribution, which, in saline lacustrine source rocks, has a maximum peak at $C_{23}$ and, in freshwater lacustrine source rocks, has a maximum peak at $C_{21}$. Therefore, $C_{21}/C_{23}$ tricyclic terpane ratios can be used to reflect changes in the salinity of the water column [34,35]. In this study, the values of $C_{21}/C_{23}$ tricyclic terpane ratios in source rock samples from LS2 range from 0.86 to 3.00, with a mean of 1.67, which indicates that they are deposited in a fresh water to brackish environment.

Abundant $C_{24}$ 17, 21-secohopane ($C_{24}$ Tet), detected in mass chromatogram using m/z 191, may indicate a fresh water to brackish environment [21,34–36]. It is commonly expressed as $C_{24}$ Tet/$C_{26}$ TT to reflect changes in the salinity of the water column. The values of $C_{24}$ Tet/$C_{26}$ TT ratios in source rock samples from LS2 range from 0.97 to 2.42, with a mean of 1.58.

### 5.5. Dynamical Formation Model of the Lacustrine OM

According to the above comprehensive analysis, we concluded that, during the deposition of organic-rich shales in LS2, a oxic-suboxic and fresh to brackish environment

was experienced, with a relatively severe algae bloom, which may be the joint result of the warm and humid paleo-climate, moderate chemical weathering, and an abundance of nutrients inputted into the water body of the paleo-lake [7]. The sedimentation and the subsequent decomposition of OM led to oxygen and other oxidant depletion and, concomitantly, promoted the remineralization of phosphorus and other nutrient elements, which may result in phosphorus and other nutrients released into the water body and causing paleo-productivity to increase further (Figure 11) [37–42].

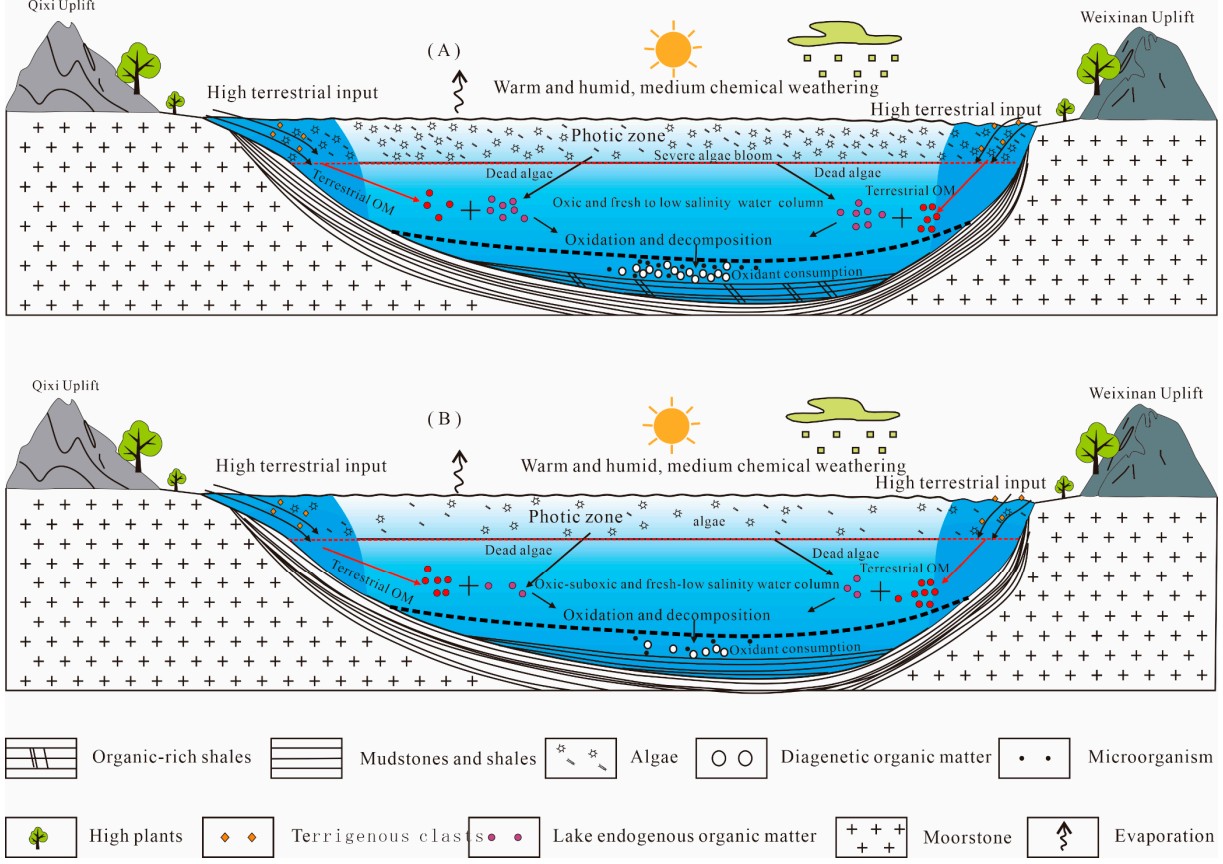

**Figure 11.** Conceptual diagram along the basin illustrating the factors that influenced the OM enrichment during LS2 of the Weixinan Sag, Beibuwan Basin. (**A**) A high-productivity and oxic to suboxic dynamical formation model, based on the mechanism for the enrichment of OM in organic-rich shales (Model A); (**B**) low-productivity and oxic to sub-oxic dynamical formation model (Model B), based on the mechanism for the enrichment of OM in mudstones and shales.

The mudstones/shales were deposited In similar redox and salinity conditions, with weaker algae blooms compared with that of organic-rich shales, which may be caused by the relatively few nutrients provided owing to the parent rocks of the mudstone/shales being acidic granite (Figure 11) [7].

Taking together, the different paleo-productivity conditions of the overlaying water layers played a main role in the different degrees of organic matter enrichment between the organic-rich shales and mudstones/shales from LS2 of the Weixinan Sag, Beibuwan Basin.

## 6. Conclusions

Rock-Eval pyrolysis data suggest that the OM of organic-rich shales from LS2 is type I and II kerogen with low maturity, and that in mudstones/shales is type II kerogen with similar maturity compared with that of organic-rich shales. The distributions of biomarkers (*n*-alkanes and regular steranes) indicate a mainly lake algae and aquatic macrophytesl origin in organic-rich shales from LS2, and the OM of mudstones/shales is mainly contributed from terrestrial higher plants.

Moreover, LS2 was deposited in a fresh to low salinity water column, in oxic to sub-oxic environments, indicated by the relatively low gammacerane index and $C_{24}$ Tet/$C_{26}$ TT, high Pr/Ph ratios, and $C_{21}$/$C_{23}$ TT, and confirmed by a cross-plot of $C_{21–22}$/$C_{27–29}$ sterane versus dia/reg $C_{27}$ sterane ratios.

The values of the 4MSI in organic-rich shales from LS2 were relatively higher than those of the mudstones/shales, which suggest that the paleo-lake experienced more severe algal blooms during the sedimentation of the organic-rich shales compared with during the deposition of the mudstones/shales. Based on this investigation, two dynamical formation models are proposed: a high-productivity and oxic to sub-oxic dynamical formation model (Model A) and a low-productivity and oxic to sub-oxic dynamical formation model (Model B).

**Author Contributions:** X.Y.: Conceptualization, Investigation, Formal analysis, Methodology and Writing—original draft; X.L.: Formal analysis, Validation, writing—review & editing and Software; Y.H. (Yahao Huang): Resources, Writing—review & editing, Validation and Software; Y.H. (Yulong He): Writing—review & editing; R.Y.: Visualization and Writing—review & editing; R.W.: Writing—review & editing; P.P.: Writing—review & editing. All authors have read and agreed to the published version of the manuscript.

**Funding:** This research was funded by Petrochina Carbonate Reservoir Key Laboratory Foundation, grant number: RIPED-2022-JS-2382.

**Data Availability Statement:** The data used to support the findings of this study are included within the article.

**Conflicts of Interest:** The authors declare no conflict of interest.

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
