# Peer review of "The Depositional Environment of the Lacustrine Source Rocks in the Eocene Middle Number of the Liushagang Formation of the Weixinan Sag, Beibuwan Basin, China: Implications from Organic Geochemical Analyses"

_minerals, doi:10.3390/min13040575_

Round 1
Reviewer 1 Report
This manuscript investigated the depositional environment of the middle number of Liushagang Formation source rocks from the Weixinan Sag, Beibuwan Basin. The genetics of the type of kerogen, thermal maturity, origin, paleo-productivity and depositional environment of organic matters were analyzed. Overall, this work is well-organised and potentially published. The topic is of broader interest to Minerals readers. When reading the manuscript, I have a few points that hopefully can help improve it.
(1) This is a good case study, but the methods are commonly used, and the conclusions are not new for the scientific world, suggesting that the innovation of the work is insufficient.
(2) More discussion on the source and depositional environment of organic matter in different type of source rocks is needed.
(3) Different sedimentary models for organic-rich shales and mudstone/shales were build, but lack of sufficient evidence.
(4) Lines 215-216: Are there any mistakes in the values of GI.
(5) The distribution of n-alkanes was also useful for indicating origin.
(6) The ternary plot of C27-C29 regular steranes is commonly used to indicate origin.
(7) What is the significance of the C21TT/C23TT in this study?
(8) Line 250: “the OM of organic-rich shales is Type I and II1 kerogen and of 250 mudstones/shales is Type II2 and â…¢ kerogen”, inaccurate description of type of kerogen.
(9) Line 274: What is the unique depositional environment of LS2? Please clarify it.
(10) There are many spell mistakes, for example, the “n” should be italic, Fig. 7: Posmature should be replaced by Postmature.
(11) The English expression needs to be improved.
Author Response
Reviewer #1
Comments and Suggestions for Authors
This manuscript investigated the depositional environment of the middle number of Liushagang Formation source rocks from the Weixinan Sag, Beibuwan Basin. The genetics of the type of kerogen, thermal maturity, origin, paleo-productivity and depositional environment of organic matters were analyzed. Overall, this work is well-organised and potentially published. The topic is of broader interest to Minerals readers. When reading the manuscript, I have a few points that hopefully can help improve it.
- This is a good case study, but the methods are commonly used, and the conclusions are not new for the scientific world, suggesting that the innovation of the work is insufficient.
Response 1: Thanks for your affirmation of our work.
(2) More discussion on the source and depositional environment of organic matter in different type of source rocks is needed.
Response 2: Thanks; Based on answering comments of 5, 6 and 7, we added several discussions on the source and depositional environment of organic matter in different type of source rocks.
(3) Different sedimentary models for organic-rich shales and mudstone/shales were build, but lack of sufficient evidence.
Response 3: Thanks, and we added researches on inorganic geochemistry according to reference (Cao et al., 2020) to make the evidence more sufficient in building different sedimentary models for organic-rich shales and mudstone/shales.
(4) Lines 215-216: Are there any mistakes in the values of GI.
Response 4: Thanks, and we corrected mistakes in the values of GI in article.
(5) The distribution of n-alkanes was also useful for indicating origin.
Response 5: Thanks, and we added the discussions of distribution of n-alkanes to illustrate the origin of organic matter in article.
(6) The ternary plot of C27-C29 regular steranes is commonly used to indicate origin.
Response 6: Thanks, and previous studies showed that the ratios of C27/C29 regular steranes can be used to indicate origin of organic matter simply and clearly (Bao et al., 2007), while there is uncertainty in the ternary plot of C27-C29 regular steranes used to indicate origin. Therefore, we selected the the ratios of C27/C29 regular steranes to reflect origin of organic matter in this study.
(7) What is the significance of the C21TT/C23TT in this study?
Response 7: Thanks; The C21 TT/C23 TT ratios can be used to reflect changes in salinity of water column (Bao et al., 2018), the analyses of which were added in this study in article.
(8) Line 250: “the OM of organic-rich shales is Type I and II1 kerogen and of 250 mudstones/shales is Type II2 and â…¢ kerogen”, inaccurate description of type of kerogen.
Response 8: Thanks; We corrected the type of kerogen: “the OM of organic-rich shales is Type I and II kerogen and that of mudstones/shales is Type II kerogen”.
(9) Line 274: What is the unique depositional environment of LS2? Please clarify it.
Response 9: Thanks; The unique depositional environment of LS2 is a fresh to low salinity water column, with oxic-suboxic environments.
(10) There are many spell mistakes, for example, the “n” should be italic, Fig. 7: Posmature should be replaced by Postmature.
Response 10: Thanks, and we corrected the word spell mistakes in detail in article.
(11) The English expression needs to be improved.
Response 11: Thanks; We had improved the English expression in detail.
- Cao, L.;Zhang, Z.H; Li, H.Y; Zhong, N.N; Xiao, L.L; Jin, X.; Li, H. Mechanism for the enrichment of organic matter in the Liushagang Formation of the Weixinan Sag, Beibuwan Basin, China. Petrol. Geol. 2020, 122, 104649.
- Bao, J.P.; Zhu, C.S.; Ni, C.H. Distribution and composition of biomarkers in crude oils fromdifferent sags of Beibuwan Basin. Acta Sedimentologica Sinica. 2007, 25, 646-652 (In Chinese with English abstract).
- Bao,J.P.; Wang, Z.F.; Zhu C.S.; Wang, L.Q.; Chen, Y.; Zhou, F. A new kind of crude oils and the geochemical characteristics in the Dongping Area,Qaidam Basin. Acta Sedimentologica Sinica. 2018, 36, 829-841 (In Chinese with English abstract).

Reviewer 2 Report
REVIEW
to the article by Xiaoyong Yang , Xiaoxia Lv, Yahao Huang, Rui Yang, Ruyue Wang and Peng Peng
"Depositional environment of the lacustrine source rocks in Middle number of Liushagang Formation of the Weixinan Sag, Beibuwan Basin, China: Implications from organic geochemical analyses"
submitted to Minerals
The article is devoted to the study of the chemical composition of organic matter from shales of the middle part of the Liushagang Formation (LS2) of the Weixinan Sag, Beibuwan Basin. Based on this analysis, two dynamical formation models were proposed: a high-productivity and oxic-suboxic dynamical formation model (Model I) and a low-productivity and oxic-suboxic dynamical formation model (Model II).
In general, the article can be published in the journal Minerals, although it is a narrow geochemical study that is aimed at finding oil and gas deposits.
Key notes:
There is no problem in the introduction, the purpose and objectives of which the article is aimed at solving.
Line 92. Figure 1 needs to be redone. Make the colored part of the picture larger, and cut off the excess. Indicate the name of the cities as on geographical maps.
Line 103. In Figure 2, there is no green in the legend. The value of lithological thicknesses is not indicated in meters. Organic shales (argillites) are poorly described in the text from the point of view of geology. The composition of clay minerals is not specified. In the drawings they should be shown in black, as the coals show.
Line 168. Tables one and two are overloaded. It needs to be simplified or replaced with graphs.
Line 292. Wrong literature reference. “In addition, according to Wang et al. (2015), pregnane and its homologues are related 292 with restricted, clastic-starved marine or saline deposition settings[26]”.
Line 330. Figure 10. Conceptual diagram…
1) What is “medical chemical weathering?” In geology, there are two types of weathering: chemical and physical. What is medicine here? What do the red circles and diamonds mean?
Author Response
Reviewer #2
Comments and Suggestions for Authors
REVIEW to the article by Xiaoyong Yang , Xiaoxia Lv, Yahao Huang, Rui Yang, Ruyue Wang and Peng Peng "Depositional environment of the lacustrine source rocks in Middle number of Liushagang Formation of the Weixinan Sag, Beibuwan Basin, China: Implications from organic geochemical analyses" submitted to Minerals
The article is devoted to the study of the chemical composition of organic matter from shales of the middle part of the Liushagang Formation (LS2) of the Weixinan Sag, Beibuwan Basin. Based on this analysis, two dynamical formation models were proposed: a high-productivity and oxic-suboxic dynamical formation model (Model I) and a low-productivity and oxic-suboxic dynamical formation model (Model II).
In general, the article can be published in the journal Minerals, although it is a narrow geochemical study that is aimed at finding oil and gas deposits.
Key notes:
There is no problem in the introduction, the purpose and objectives of which the article is aimed at solving.
- Line 92.Figure 1 needs to be redone. Make the colored part of the picture larger, and cut off the excess. Indicate the name of the cities as on geographical maps.
Response 1: Thanks, and all done as suggested.
- Line 103.In Figure 2, there is no green in the legend. The value of lithological thicknesses is not indicated in meters. Organic shales (argillites) are poorly described in the text from the point of view of geology. The composition of clay minerals is not specified. In the drawings they should be shown in black, as the coals show.
Response 2: Thanks. (1) We changed the “green” to “black” and added it into legends in the bottom of Fig. 3; (2) The value of lithological thicknesses indicated in meters was added into Fig. 3; (3) we added the detailed description of organic shales (argillites) into article; (4) the description of composition of clay minerals was added into article (5) done as suggested.
- Line 168.Tables one and two are overloaded. It needs to be simplified or replaced with graphs.
Response 3: Thanks, and done as suggested.
- Line 292.Wrong literature reference. “In addition, according to Wang et al. (2015), pregnane and its homologues are related 292 with restricted, clastic-starved marine or saline deposition settings[26]”.
Response 4: Thanks; we corrected the wrong literature reference.
- Line 330.Figure 10. Conceptual diagram…
- What is “medical chemical weathering?” In geology, there are two types of weathering: chemical and physical. What is medicine here? What do the red circles and diamonds mean?
Response 5: Corrected, and we had changed the “medical” to “medium” in article. Thanks, the red circles mean terrigenous clasts and the red diamonds are represent as terrigenous organic matter; and we added the red circles and diamonds into legends in the bottom of Fig. 11.

Reviewer 3 Report
Please see my comments on the PdF file. You have to improve maps and you have to draw some new sections.

Author Response
Reviewer #3
Comments and Suggestions for Authors
Please see my comments on the PDF file. You have to improve maps and you have to draw some new sections.
- What is the Middle number. he title must be representative of what you present. There is no any use of such numbers in your stratigraphic column or in your map. As I think that your work was focused on Eocene deposits you have to put this age in the title
Response 1: Thanks. (1) Previous studies showed that the Eocene Liushagang Formation is divided into three members, designated as LS1 on the top, LS2 in the middle and LS3 at the bottom, and we added the use of such numbers in the stratigraphic column; (2) Done as suggested: we putted the “Eocene” in the title.
- Capital C
Response 2: Thanks, and done as suggested.
- As this is the representative map I would like to see a more detailed geological map in order to show the details of what you study. I suggest either to complete the b map with geological - sedimentological informations or to add anew one.
Response 3: Thanks, and done as suggested.
- Moreover, as this map must support your figure 10 you have to draw two geological cross-sections in order to show the depositional thickness and the distribution of the sediments along and across the basin, in order to support the restriction.
Response 4: Thanks, and done as suggested. We added a geological cross-section to show the depositional thickness, and the distribution of the sediments along and across the basin showed in Fig. 1b.
- According this part of the map it seems that the Haizhong sag was connected with the Weixinan sag. It is true? or not?
Response 5: Thanks. Previous studies suggested that the Weixinan sag and the Haizhong sag was locally connected in the Paleocene, and fully connected in the Eocene and Oligocene.
- What fault? normal? used the international symbols for normal or reverse faults. If these are normal faults then we are waiting to have some throughs and some highs. According your stratigraphic column it seems that these faults could be normal during Eocene and the inversion strarted during Miocene and so on during Pliocene these faults are thrust faults? These faults are active from when and until when?
Response 6: Thanks; (1) These faults are normal faults; (2) These faults are active from Upper Cretaceous to Pliocene.
- As these stages are very critical in rder to undestand the basin evolution you must draw and some evolutionary sections showing the inversion.
Response 7: Thanks, and done as suggested.
- These sections are along the basin or across the basin? Probably across the basin. In this case you have to add also the outcrop relief in order to show the reason of terrestrial input.
Response 8: Thanks. (1) These sections are along the basin; (2) Done as suggested.
- Probably this model represent the Eocene conditions, as the above restriction was during Eocene. Explain either in the text or in the caption. When you will read my comments on figures 1 and 2 you will understand the conditions.
Response 9: Thanks for your suggestion, and we agree with your opinion.

Round 2
Reviewer 2 Report
Line 374. The last drawing. Redo the legend and remove it twice (repeats) Diagenics... . The inscription on the left High terris... do it horizontal. What does +++ mean? What do the purple circles mean? From figure It is not clear what is the fundamental difference between the upper model and the lower one. Make the characters bigger. Taller plants should be replaced trees instead of showing with arrows (see the model in the article https://doi.org/10.1007/s12040-021-01771-3).
Author Response
Response 1: Thanks. (1) Done as suggested; (2) Done as suggested; (3) The +++ means granite and the purple circles mean lake endogenous organic matter; (4) Done as suggested and we had quoted this article (https://doi.org/10.1007/s12040-021-01771-3) for reference.
